# Bcr4 Is a Chaperone for the Inner Rod Protein in the *Bordetella* Type III Secretion System

Masataka Goto,[a] Akio Abe,[a] Tomoko Hanawa,[b] Masato Suzuki,[c] ⓘ Asaomi Kuwae[a]

[a]Laboratory of Bacterial Infection, Graduate School of Infection Control Sciences, Kitasato University, Tokyo, Japan
[b]Department of Infectious Diseases, Kyorin University School of Medicine, Tokyo, Japan
[c]Antimicrobial Resistance Research Center, National Institute of Infectious Diseases, Tokyo, Japan

**ABSTRACT** *Bordetella bronchiseptica* injects virulence proteins called effectors into host cells via a type III secretion system (T3SS) conserved among many Gram-negative bacteria. Small proteins called chaperones are required to stabilize some T3SS components or localize them to the T3SS machinery. In a previous study, we identified a chaperone-like protein named Bcr4 that regulates T3SS activity in *B. bronchiseptica*. Bcr4 does not show strong sequence similarity to well-studied T3SS proteins of other bacteria, and its function remains to be elucidated. Here, we investigated the mechanism by which Bcr4 controls T3SS activity. A pulldown assay revealed that Bcr4 interacts with BscI, based on its homology to other bacterial proteins, to be an inner rod protein of the T3SS machinery. An additional pulldown assay using truncated Bcr4 derivatives and secretion profiles of *B. bronchiseptica* producing truncated Bcr4 derivatives showed that the Bcr4 C-terminal region is necessary for the interaction with BscI and activation of the T3SS. Moreover, the deletion of BscI abolished the secretion of type III secreted proteins from *B. bronchiseptica* and the translocation of a cytotoxic effector into cultured mammalian cells. Finally, we show that BscI is unstable in the absence of Bcr4. These results suggest that Bcr4 supports the construction of the T3SS machinery by stabilizing BscI. This is the first demonstration of a chaperone for the T3SS inner rod protein among the virulence bacteria possessing the T3SS.

**IMPORTANCE** The type III secretion system (T3SS) is a needle-like complex that projects outward from bacterial cells. *Bordetella bronchiseptica* uses the T3SS to inject virulence proteins into host cells. Our previous study reported that a protein named Bcr4 is essential for the secretion of virulence proteins from *B. bronchiseptica* bacterial cells and delivery through the T3SS. Because other bacteria lack a Bcr4 homologue, the function of Bcr4 has not been elucidated. In this study, we discovered that Bcr4 interacts with BscI, a component of the T3SS machinery. We show that a *B. bronchiseptica* BscI-deficient strain was unable to secrete type III secreted proteins. Furthermore, in a *B. bronchiseptica* strain that overproduces T3SS component proteins, Bcr4 is required to maintain BscI in bacterial cells. These results suggest that Bcr4 stabilizes BscI to allow construction of the T3SS in *B. bronchiseptica*.

**KEYWORDS** Bcr4, *Bordetella*, BscI, T3SS, chaperone, inner rod

The genus *Bordetella* consists of Gram-negative bacteria that infect the respiratory tracts of mammals, including humans. *Bordetella pertussis* causes a severe coughing attack called whooping cough in humans (1, 2). *Bordetella bronchiseptica* causes atrophic rhinitis in pigs and kennel cough in dogs (1, 2). These *Bordetella* spp. harbor a virulence factor secretion apparatus called the type III secretion system (T3SS).

The T3SS consists of a basal body that penetrates the inner and outer membranes of the bacteria and a needle structure that protrudes outside the bacteria, and is

**Ad Hoc Peer Reviewer** ⓘ Zongmin Du, Beijing Institute of Microbiology and Epidemiology; Julien Bergeron, King's College London

Address correspondence to Asaomi Kuwae, kuwae@lisci.kitasato-u.ac.jp.

The authors declare no conflict of interest.

conserved in many Gram-negative bacteria such as *Yersinia*, *Salmonella*, and *Pseudomonas*. *B. bronchiseptica* uses the T3SS to inject virulence factors, called effectors, into host cells to disrupt the physiological functions of host cells (3). Once the basal body and the export apparatus of the T3SS are completed, the type III secreted proteins are secreted in a fixed order. First, the components (see Table S1 in the supplemental material) of the needle structure (SctF) and the inner rod (SctI) that ties the needle structure to the basal body are secreted. After the needle is completed, translocators (SctE and SctB) are secreted, which form small pores in the host cell membrane to create a pathway for effectors. Finally, the effectors translocate into the host cell via the T3SS (4, 5). In *B. bronchiseptica*, BopB (SctE) (6), BopD (SctB) (7), and Bsp22 (SctA) (8) function as translocators (see Table S1), while BteA (also referred to as BopC) (9, 10), BopN (11, 12), and BspR (also referred to as BtrA) (13, 14) function as effectors. Bsp22 is located at the tip of the needle and bridges the needle to the pore-forming factors embedded in the plasma membrane (15). BteA has been shown to cause membrane-disrupting cytotoxicity in mammalian cells (10). BspR is a regulator that represses transcription of the *bteA* gene and genes on the *bsc* locus (see Fig. S1), where genes encoding type III secreted proteins and components of T3SS are located (14, 16). According to secondary structure prediction—e.g., the predicted positions of the helix and the overall structure of the operon—the BscF and BscI of *B. bronchiseptica* correspond to *Yersinia* needle YscF (SctF) and inner rod YscI (SctI), respectively (see Table S1).

In addition, many type III secreted proteins have unique chaperones that are involved in stabilizing the substrate and preventing premature polymerization of the substrate in the bacterial cytosol, and then in efficiently transporting the substrate to the T3SS machinery (17–20). For example, PscE and PscG function as chaperones of the needle PscF (SctF) in *Pseudomonas* (see Table S1). These chaperones stabilize PscF in the bacterial cytosol and are involved in PscF secretion through the T3SS (21). In bacteria such as *Yersinia* and *Pseudomonas*, an inner rod chaperone is thought to exist, because the inner rod is secreted out of the bacterial cell and polymerized (22, 23). However, a chaperone for the inner rod has not been reported.

Thus far, a chaperone-like protein called Bcr4 has been identified in *B. bronchiseptica* (24). *Bordetella* Bcr4 is highly conserved among *B. pertussis*, *B. parapertussis*, and *B. bronchiseptica* (see Fig. S2). In the present study, we attempted to identify factors that interact with Bcr4 in order to investigate how Bcr4 is involved in the T3SS regulation.

## RESULTS

**Bcr4 binds to BscI, an inner rod protein of the *Bordetella* type III secretion system.** The results of a previous study suggested that Bcr4 is a chaperone for components of the T3SS (24). T3SS chaperones are known to be involved in substrate stability and efficient transport of substrates to the T3SS machinery (17, 19, 20). In addition, it is generally known that the genes of these chaperones are localized adjacent to genes encoding their substrates (25). On the *B. bronchiseptica* S798 chromosome, the genes encoding BcrH2, BscI, BscJ, and BscK are located in the vicinity of the *bcr4* gene (Fig. 1A), and these proteins are predicted to function as a translocator chaperone, inner rod (SctI), inner membrane ring (SctJ), and ATPase cofactor (SctK), respectively (see Table S1) (5, 26). To test whether Bcr4 binds to BscI, BscJ, or BscK, we added *Escherichia coli* lysates containing the V5-tagged target proteins (BscI-V5, BscJ-V5, or BscK-V5) to Strep-Tactin beads loaded with the purified Strep-tagged Bcr4 (Bcr4-Strep) and then performed the pulldown assay. The supernatant fraction (Sup) and pellet fraction (Pellet) samples were prepared, separated by SDS-PAGE, and subjected to Western blotting with anti-V5 antibody (Fig. 1B). When the beads were washed with Tris-buffered saline (TBS), the V5 signal of the BscI-V5 pellet sample was detected in the beads loaded with Bcr4 but not in the unloaded beads (Fig. 1B). The V5 signals of the BscK-V5 and BscJ-V5 pellet samples were detected in both the Bcr4-loaded beads and the unloaded beads when washed with TBS and were not detected when washed with TBS containing 0.1% Triton X-100 (Fig. 1B). These results suggest that Bcr4 binds to BscI, an inner rod protein of the *Bordetella* T3SS.

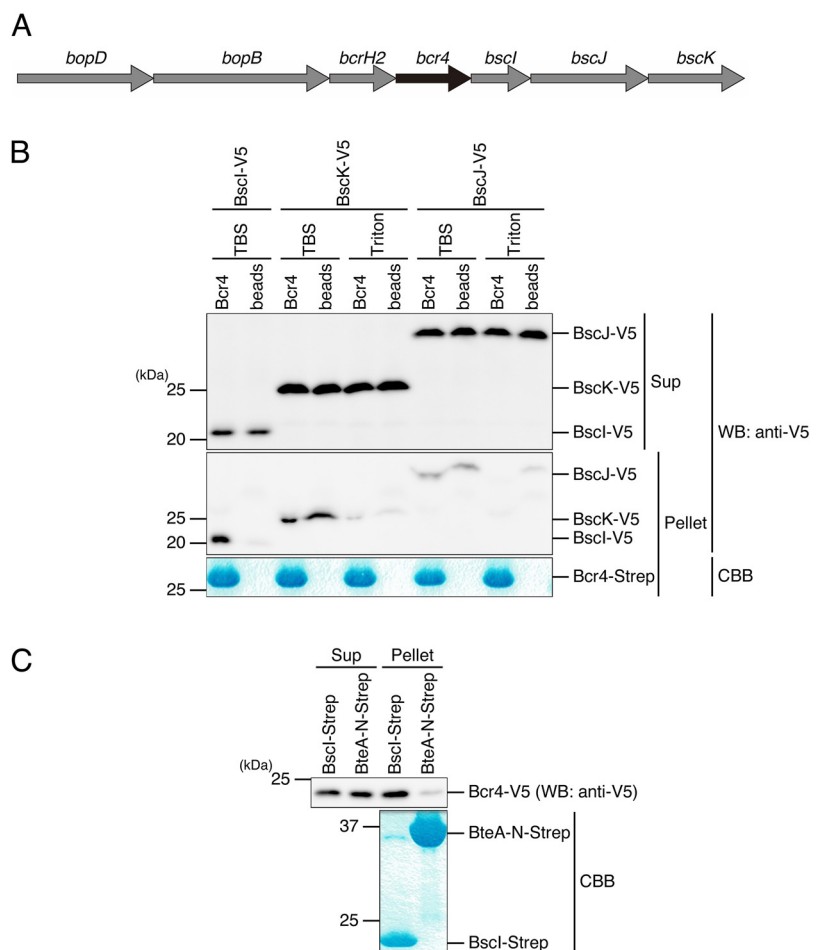

**FIG 1** Interaction of Bcr4 with inner rod protein BscI. (A) The *bcr4* gene and its peripheral genes localized in the T3SS apparatus locus (*bsc* locus) on the *B. bronchiseptica* S798 chromosome are depicted. (B) The purified Bcr4 (Bcr4-Strep) was loaded on Strep-Tactin beads. Then, the Bcr4-Strep-loaded beads (Bcr4) and the beads alone (beads) were mixed with lysates prepared from *E. coli* BL21 producing BscI, BscK, or BscJ tagged with V5 (BscI-V5, BscK-V5, or BscJ-V5), respectively. After 3 h of incubation at 4°C, each supernatant was prepared as the supernatant fraction (Sup) sample, and each pellet was washed with TBS alone or TBS containing 0.1% Triton X-100 (Triton) and prepared as the pellet fraction (Pellet) sample. The Sup and Pellet samples were separated by SDS-PAGE and analyzed by Western blotting (WB) with anti-V5 antibody (top). Pellet samples were also stained with Coomassie brilliant blue (CBB, bottom). (C) The purified BscI (BscI-Strep) or N-terminal moiety of BteA (amino acid region 1–312, BteA-N-Strep) was loaded on Strep-Tactin beads. The beads were then mixed with the lysate prepared from *E. coli* BL21 to produce Bcr4 tagged with V5 (Bcr4-V5) at 4°C for 3 h. The prepared Sup and Pellet samples were separated by SDS-PAGE and analyzed by WB with anti-V5 antibody (top). Pellet samples were also stained with CBB (bottom). Experiments were performed at least three times, and representative data are shown.

Next, to confirm that Bcr4 binds to BscI, a pulldown assay was performed by adding *E. coli* lysate containing V5-tagged Bcr4 (Bcr4-V5) to Strep-Tactin beads with the purified Strep-tagged BscI (BscI-Strep) or BteA N-terminal amino acid region 1–312 (i.e., the region from amino acid 1 to amino acid 312; BteA-N-Strep). BteA is a protein secreted from the T3SS and interacts with its cognate chaperone BtcA through the N-terminal (9). As a result, the V5 signal was detected in the pellet sample of the BscI-Strep but was hardly detected in that of the BteA-N-Strep (Fig. 1C). These results strongly suggested that Bcr4 binds to BscI.

**The C-terminal region of Bcr4 is required for the binding of Bcr4 to BscI.** Next, to investigate the Bcr4 region responsible for the binding to BscI, we produced Strep-tagged full-length Bcr4 (Bcr4-FL-Strep), Bcr4 lacking amino acid region 58–109 (Bcr4Δ58-109-Strep), and Bcr4 lacking amino acid region 110–173 (Bcr4Δ110-173-Strep) in *E. coli* (Fig. 2A). A pulldown assay was performed by adding *E. coli* lysate containing V5-tagged BscI (BscI-V5) to

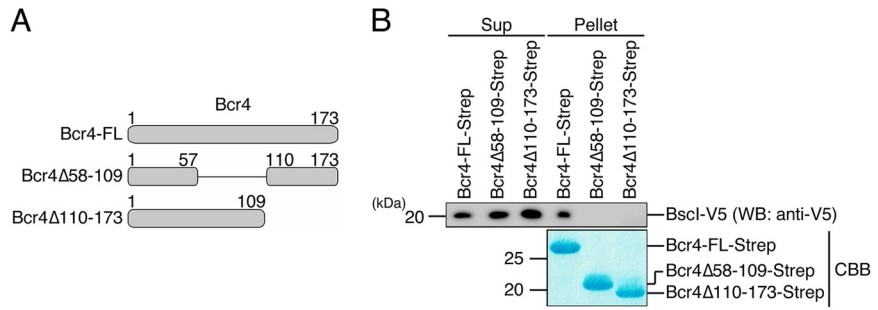

**FIG 2** Pulldown assays between BscI and truncated versions of Bcr4. (A) Bcr4 derivatives used for pulldown assay are depicted. (B) The purified Bcr4-FL, Bcr4Δ58-109, or Bcr4Δ110-173 (Bcr4-FL-Strep, Bcr4Δ58-109-Strep, or Bcr4Δ110-17-Strep) was loaded on Strep-Tactin beads, and the beads were then mixed with the lysate prepared from *E. coli* BL21, producing BscI-V5, respectively. After 3 h of incubation at 4°C, the supernatant fraction (Sup) and pellet fraction (Pellet) samples were prepared. The Sup and Pellet samples were separated by SDS-PAGE and analyzed by WB with anti-V5 antibody (top), and stained with CBB (bottom). Experiments were performed at least three times, and representative data are shown.

Strep-Tactin beads with the purified Bcr4-FL-Strep, Bcr4Δ58-109-Strep, or Bcr4Δ110-173-Strep. The Sup and Pellet samples were then prepared, separated by SDS-PAGE, and subjected to Western blotting using anti-V5 antibody (Fig. 2B). As a result, the V5 signal was detected in the pellet sample of Bcr4-FL-Strep, but not in those of Bcr4Δ58-109-Strep and Bcr4Δ110-173-Strep. Although it is still unknown which region of Bcr4 directly interacts with BscI, our results strongly suggest that both the Bcr4-58-109 and Bcr4-110-173 regions are required for the interaction.

**The C-terminal region of Bcr4 is required for T3SS activity.** The results presented in Fig. 2 suggest that the C-terminal region of Bcr4 is required for binding to BscI. Therefore, we introduced plasmids encoding the full-length FLAG-tagged Bcr4 (Bcr4-FL-FLAG), Bcr4 with 5 C-terminal amino acids deleted (Bcr4Δ169-173-FLAG), Bcr4 with the 10 C-terminal amino acids deleted (Bcr4Δ164-173-FLAG), and Bcr4 with the 15 C-terminal amino acids deleted (Bcr4Δ159-173-FLAG) (Fig. 3A) into a Bcr4-deficient strain (Δ*bcr4*) (Δ*bcr4*+*bcr4*-FL-FLAG, Δ*bcr4*+*bcr4*Δ169-173-FLAG, Δ*bcr4*+*bcr4*Δ164-173-FLAG, and Δ*bcr4*+*bcr4*Δ159-173-FLAG, respectively). Whole-cell lysate (WCL) samples and culture supernatant fraction (CS) samples were prepared from these strains and separated by SDS-PAGE. Western blotting was then carried out with antibodies against Bcr4, FLAG, BteA (an effector and a type III secreted protein), BopD (SctB, a translocator and a type III secreted protein), or RpoB (an internal control of WCL) (Fig. 3B). Since the antibody against Bcr4 was generated using its 18 C-terminal amino acids (amino acids 156 to 173) as the antigen peptide, we considered that it may not recognize the Bcr4 partial deletion mutant proteins used in this experiment. Therefore, we also performed Western blotting with anti-FLAG antibody (Fig. 3B). As a result, when anti-Bcr4 antibody was used against the WCL samples, signals were detected in the wild-type, Δ*bcr4*+*bcr4*-FL-FLAG, and Δ*bcr4*+*bcr4* Δ169-173-FLAG strains, but almost no signals were detected in Δ*bcr4*+*bcr4*Δ164-173-FLAG and Δ*bcr4*+*bcr4*Δ159-173-FLAG strains (Fig. 3B). On the other hand, when anti-FLAG antibody was used, signals were detected in Δ*bcr4*+*bcr4*Δ164-173-FLAG and Δ*bcr4*+*bcr4*Δ159-173-FLAG strains (Fig. 3B). These results confirmed that the Bcr4 partial deletion mutant proteins were produced in the Δ*bcr4* strain. When anti-BteA or anti-BopD antibodies were used against the CS samples, signals were detected in the wild-type, Δ*bcr4*+*bcr4*-FL-FLAG, Δ*bcr4*+*bcr4*Δ169-173-FLAG and Δ*bcr4*+*bcr4*Δ164-173-FLAG, respectively, but not in Δ*bcr4* or Δ*bcr4*+*bcr4*Δ159-173-FLAG (Fig. 3B). We attempted to create *B. bronchiseptica* strains that produce shorter Bcr4, e.g., amino acid regions 1–57, 58–109, and 110–173; however, these truncated Bcr4s were produced at very low levels (data not shown). Therefore, we were unable to evaluate whether or not these truncated proteins were functional in *B. bronchiseptica*. We then infected L2 cells (a rat lung epithelial cell line) with a wild-type, Δ*bcr4*, Δ*bcr4*+*bcr4*-FL-FLAG, Δ*bcr4*+*bcr4*Δ169-173-FLAG, Δ*bcr4*+*bcr4*Δ164-173-FLAG, or Δ*bcr4*+*bcr4*Δ159-173-FLAG strain at a multiplicity of infection (MOI) of 50 for 1 h and measured the amounts of lactate dehydrogenase (LDH)

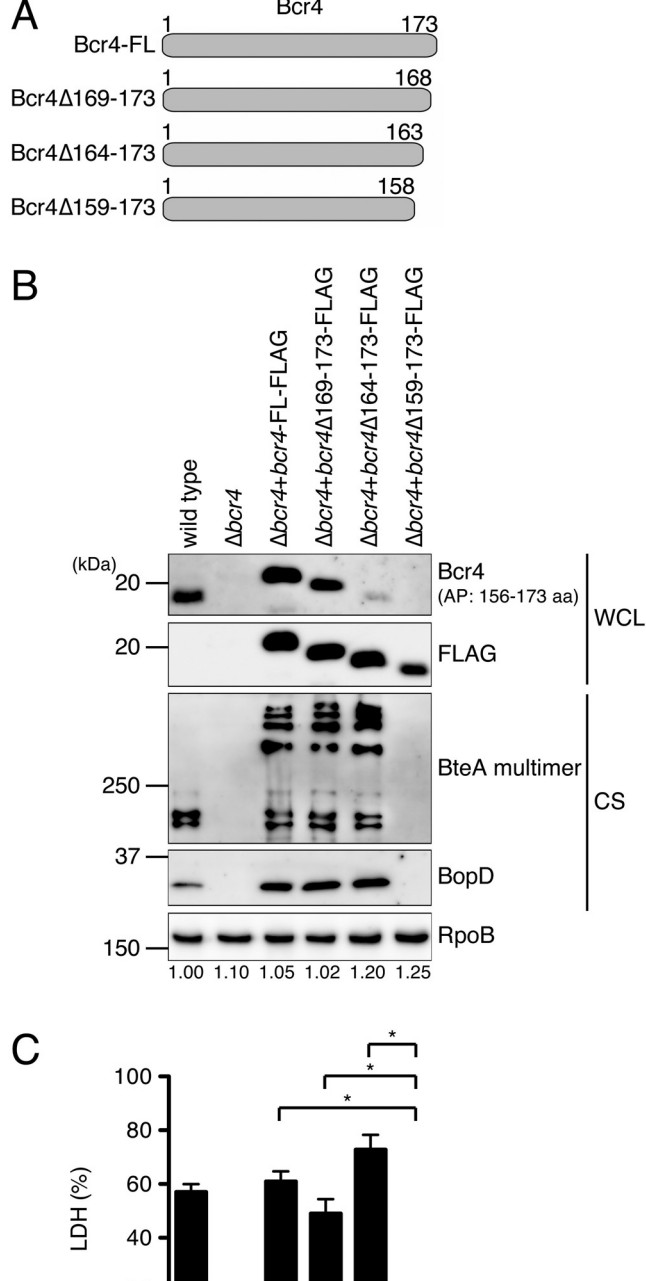

**FIG 3** The Bcr4 domain required for T3SS activity (A) Bcr4 derivatives used for the analysis of T3SS activity are depicted. (B) The *B. bronchiseptica* wild-type strain, the Bcr4-deficient strain (Δ*bcr4*), and the Δ*bcr4* strains producing Bcr4-FL-FLAG (Δ*bcr4*+*bcr4*-FL-FLAG), Bcr4Δ169-173-FLAG (Δ*bcr4*+*bcr4*Δ169-173-FLAG), Bcr4Δ164-173-FLAG (Δ*bcr4*+*bcr4* Δ164-173-FLAG), and Bcr4Δ159-173-FLAG (Δ*bcr4*+*bcr4*Δ159-173-FLAG) were cultured in SS medium. Whole-cell lysate (WCL) samples and culture supernatant fraction (CS)

released into the culture medium as an index of cytotoxicity. As a result, LDH was detected in the medium of cells infected with a Δ*bcr4*+*bcr4*-FL-FLAG, Δ*bcr4*+*bcr4*Δ169-173-FLAG, or Δ*bcr4*+*bcr4*Δ164-173-FLAG strain but not in the medium of cells infected with the Δ*bcr4*+*bcr4*Δ159-173-FLAG strain (Fig. 3C). These results suggest that the 159–163 amino acid region of Bcr4 is required for the T3SS function.

**BscI is required for the function of the T3SS in *B. bronchiseptica*.** BscI is a homologue of YscI, an inner rod (SctI) of the *Yersinia* T3SS, and YscI is essential for the function of the T3SS (22). To confirm that BscI is required for the function of the *B. bronchiseptica* T3SS, the *B. bronchiseptica* S798 wild-type, the BscI-deficient strain (Δ*bscI*), the BscI-complemented strain (Δ*bscI*+*bscI*), and the BscN-deficient strain (Δ*bscN*) were cultured in Stainer-Scholt (SS) medium. BscN is a protein predicted to function as an ATPase (SctN) and is required for the T3SS function in *Bordetella bronchiseptica* (see Table S1) (6, 27). We used the Δ*bscN* mutant as a T3SS-deficient strain. WCL and CS samples were prepared, and these samples were separated by SDS-PAGE and analyzed by Western blotting with antibodies against BscI, BteA, BopD, Bsp22 (SctA, a translocator and a component of the filament-like structure; see Fig. 7), or FHA (filamentous hemagglutinin; an adhesion factor) (Fig. 4A). As expected, the FHA signals were detected in the WCL and CS samples of all strains used here (Fig. 4A). In contrast, no signals of BscI were detected in the WCL samples of all strains used here (Fig. 4A). The BscI signal was detected in the CS of the Δ*bscI*+*bscI* mutant, and a faint BscI signal was detected in the CS of the wild type (Fig. 4A). In order to examine whether the signal we obtained at ~20 kDa by Western blotting with anti-BscI antibody (Fig. 4A) was specific or nonspecific, we prepared the supernatant fraction from the Δ*bsp22* strain. As a result, the signal disappeared in the supernatant fraction of the Δ*bsp22* strain (see Fig. S3), suggesting that the signal at ~20 kDa obtained in the Western blot represented a nonspecific interaction between anti-BscI antibody and an excess amount of Bsp22 on the membrane. The BteA, BopD, and Bsp22 signals were detected in the CS of wild-type and Δ*bscI*+*bscI* strains but not in those of Δ*bscI* and Δ*bscN* strains (Fig. 4A). BspR is a negative regulator that represses the transcription of genes on the *bsc* locus (14). In *Bordetella* strains that are unable to secrete type III secreted proteins, such as Bcr4-deficient strains (Δ*bcr4*), the repression of the *bsc* locus transcriptions by BspR is enhanced (24). Therefore, we speculated that the production of BopD and Bsp22 was intensely suppressed by BspR in the Δ*bscI* strain due to the loss of T3SS activity. To test whether this hypothesis is correct, we generated the Δ*bspR*Δ*bscI* mutant (a BspR- and BscI-deficient strain). WCL and CS samples were prepared from the *B. bronchiseptica* S798 wild-type strain and the Δ*bspR*, Δ*bspR*Δ*bscI*, and Δ*bspR*Δ*bscI*+*bscI* mutants (a BscI-complemented BspR- and BscI-deficient strain), and these samples were separated by SDS-PAGE and analyzed by Western blotting with antibodies against Bsp22 or BopB (SctE, a translocator and a type III secreted protein). As a result, the Bsp22 and BopB signals were detected in WCL samples of the wild-type, Δ*bspR*, Δ*bspR*Δ*bscI*, and Δ*bspR*Δ*bscI*+*bscI* strains (Fig. 4B). The Bsp22 and BopB signals were detected in the CS samples of the wild-type, Δ*bspR* and Δ*bspR*Δ*bscI*+*bscI* strains but not in that of the Δ*bspR*Δ*bscI* strain (Fig. 4B). These results suggest that in *B. bronchiseptica*, BscI is required for the function of the T3SS and is secreted out of the bacterial cell.

**FIG 3** Legend (Continued)

samples were separated by SDS-PAGE and analyzed by Western blotting with antibodies against Bcr4, FLAG, BteA, BopD, or RpoB. RpoB was used as an internal loading control. AP indicates the amino acids region of the antigen peptide used for anti-Bcr4 antibody generation. WCL and CS samples were prepared from equal volumes of bacterial culture. When we performed Western blotting using anti-BteA or BopD antibodies, we loaded a 100- or 50-fold smaller amount of Δ*bcr4*+*bcr4*-FL-FLAG, Δ*bcr4*+*bcr4*Δ169-173-FLAG, and Δ*bcr4*+*bcr4*Δ164-173-FLAG CS samples on the SDS-PAGE gel than on the wild-type, Δ*bcr4*, and Δ*bcr4*+*bcr4*Δ159-173-FLAG CS samples to avoid obtaining excess signal intensities. The numbers at the bottom of the lower panel indicate the relative signal intensity of RpoB measured using ImageJ software. (C) L2 cells were infected with each strain at an MOI of 50 for 1 h. The amounts of LDH released into the extracellular medium from infected cells are shown, and the relative cytotoxicity (percent) was determined as described in Materials and Methods. Error bars indicate the standard errors of the mean (SEM) from triplicate experiments. *, $P < 0.05$. Experiments were performed at least three times, and representative data are shown.

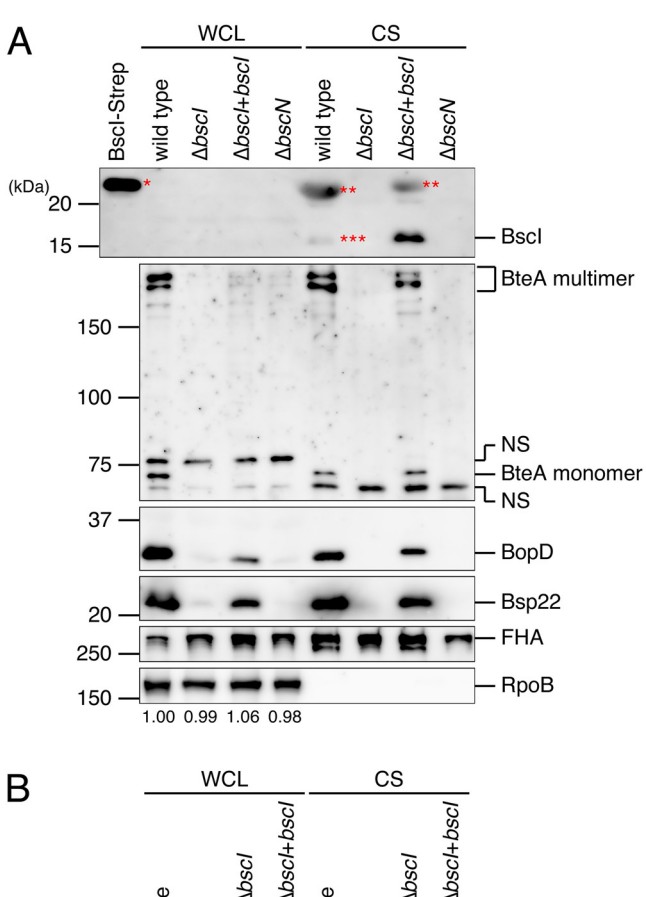

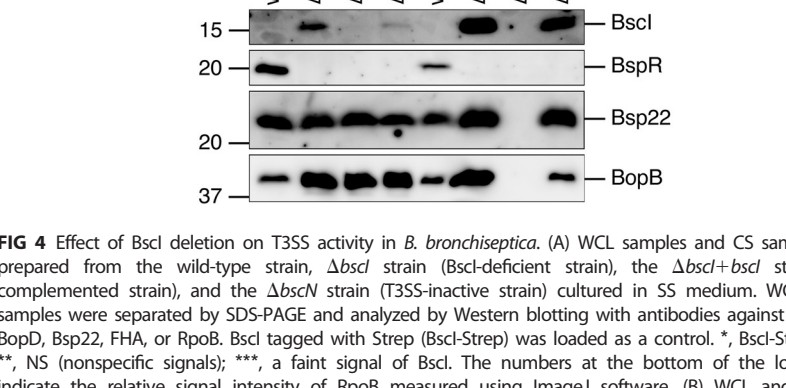

**FIG 4** Effect of BscI deletion on T3SS activity in *B. bronchiseptica*. (A) WCL samples and CS samples were prepared from the wild-type strain, Δ*bscI* strain (BscI-deficient strain), the Δ*bscI*+*bscI* strain (BscI-complemented strain), and the Δ*bscN* strain (T3SS-inactive strain) cultured in SS medium. WCL and CS samples were separated by SDS-PAGE and analyzed by Western blotting with antibodies against BscI, BteA, BopD, Bsp22, FHA, or RpoB. BscI tagged with Strep (BscI-Strep) was loaded as a control. *, BscI-Strep signal; **, NS (nonspecific signals); ***, a faint signal of BscI. The numbers at the bottom of the lower panel indicate the relative signal intensity of RpoB measured using ImageJ software. (B) WCL and CS were prepared from wild-type, Δ*bspR* (BspR-deficient strain), Δ*bspR*Δ*bscI* (BspR- and BscI-deficient strain), and Δ*bspR*Δ*bscI*+*bscI* (BscI-complemented BspR- and Bcr4-deficient strain) strains cultured in SS medium. WCL and CS were separated by SDS-PAGE and analyzed by Western blotting with antibodies against BscI, BspR, Bsp22, or BopB antibodies. Loaded WCL and CS samples were prepared from equal volumes of bacterial culture. When we performed Western blotting with antibodies against Bsp22 or BopB, we loaded a 25- or 5-fold smaller amount of Δ*bspR*, Δ*bspR*Δ*bscI*, and Δ*bspR*Δ*bscI*+*bscI* WCL samples and a 5- or 10-fold smaller amount of Δ*bspR* and Δ*bspR*Δ*bscI*+*bscI* CS samples on the SDS-PAGE gel than the wild type to avoid obtaining excess signal intensities. Experiments were performed at least three times, and representative data are shown.

**BscI plays an important role in the function of the *B. bronchiseptica* T3SS during infection of cultured mammalian cells.** To further investigate whether BscI is required for the construction and function of the T3SS in *B. bronchiseptica*, we infected L2 cells with the wild-type, Δ*bscI*, Δ*bscI*+*bscI*, or Δ*bscN* strain and measured the number of Bsp22 (SctA, a translocator, and a component of the filament-like structure; Fig. 5) signals detected on L2 cells by immunofluorescence microscopy. For the efficient

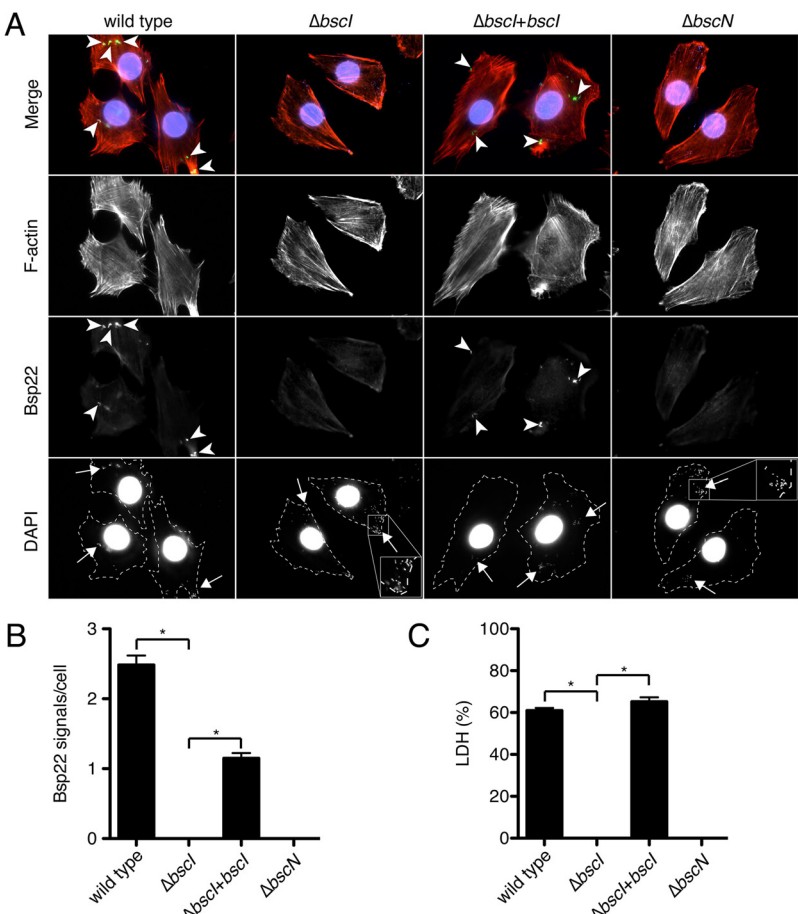

**FIG 5** Immunofluorescent staining of Bsp22 on L2 cells infected with *B. bronchiseptica* and the results of the LDH assay. (A) L2 cells were infected with the wild-type strain, the Δ*bscI* strain (BscI-deficient strain), the Δ*bscI*+*bscI* strain (BscI-complemented strain), or the Δ*bscN* strain (T3SS-inactive strain) as described in Materials and Methods. After fixation, cells were stained with anti-Bsp22 antibody (green), rhodamine phalloidin (red), and DAPI (blue). The inset shows higher magnification of the boxed area in the image. Arrowheads and arrows indicate the signals of Bsp22 (green) and bacteria, respectively. (B) The Bsp22 signals per cell were counted under a fluorescence microscope. At least 120 cells were randomly chosen. (C) The results of LDH assay at an MOI of 200 are shown as a histogram. Error bars indicate the SEMs from triplicate experiments. *, $P < 0.05$. Experiments were performed at least three times, and representative data are shown.

detection of Bsp22 signals, we cultured the bacteria in iron-depleted SS medium, which has been shown to increase the amount of type III secreted protein secreted by *B. bronchiseptica* (28). L2 cells were infected with bacteria for 1 h at an MOI of 2,000. Then, Bsp22, F-actin, and bacteria were stained with anti-Bsp22 antibody, rhodamine-phalloidin, and DAPI (4′,6′-diamidino-2-phenylindole), respectively (Fig. 5A). The DAPI fluorescence signals showing bacterial genomic DNA were detected to the same extent among the L2 cells infected with each bacterial strain, suggesting that the amounts of each mutant adhered to the cells were not significantly reduced compared to the wild-type strain (Fig. 5A). The fluorescence signals of Bsp22 were detected on L2 cells infected with the wild-type or Δ*bscI*+*bscI* strain but not on L2 cells infected with the Δ*bscI* or Δ*bscN* strain (Fig. 5A and B). We also infected L2 cells with the wild-type, Δ*bscI*, Δ*bscI*+*bscI*, or Δ*bscN* strain for 1 h at an MOI of 200 and measured the amount of LDH released into the extracellular medium. The results showed that LDH was detected in the medium of L2 cells infected with the wild-type or Δ*bscI*+*bscI* strain but not in the medium of L2 cells infected with the Δ*bscI* or Δ*bscN* strain (Fig. 5C). These results suggest that BscI is required for the construction of the T3SS and induction of T3SS-dependent death of mammalian cells in *B. bronchiseptica*.

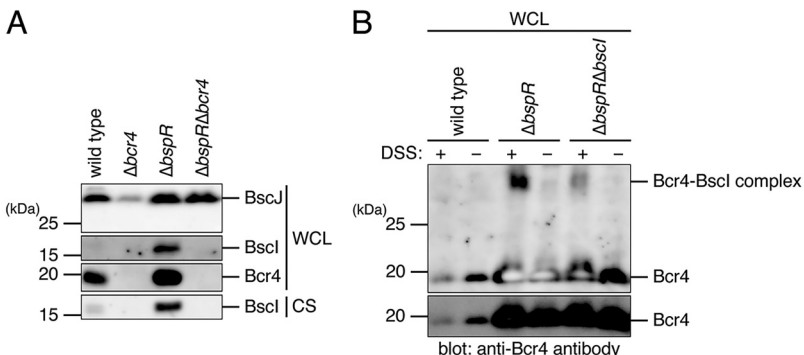

**FIG 6** The effect of Bcr4 on BscI stability and the suggestive interaction of Bcr4 with BscI in *B. bronchiseptica*. (A) The WCL samples and CS samples were prepared from the wild-type strain, the Δ*bcr4* strain (Bcr4-deficient strain), the Δ*bspR* strain (BspR-deficient strain), or the Δ*bspR*Δ*bcr4* strain (BspR- and Bcr4-deficient strain) cultured in SS medium. WCL and CS samples were separated by SDS-PAGE and analyzed by Western blotting with antibodies against BscI (inner-rod protein), BscJ (inner-ring protein), and Bcr4. The samples were prepared from equal volumes of bacterial culture. When we carried out Western blotting with anti-BscJ or Bcr4 antibodies, we loaded a 10-fold-smaller amount of Δ*bspR* or Δ*bspR*Δ*bcr4* WCL samples on the SDS-PAGE than wild-type or Δ*bcr4* WCL samples to avoid obtaining excess signal intensities. (B) WCL samples were prepared from the wild-type, Δ*bspR*, and Δ*bspR*Δ*bscI* strains treated with (+) or without (−) the cross-linker disuccinimidyl suberate (DSS). WCLs were separated by SDS-PAGE and analyzed by Western blotting with anti-Bcr4 antibody. The lower panel is a short exposure of the upper panel. Experiments were performed at least three times, and representative data are shown.

**Bcr4 is required for the stability of BscI and is suggested to interact with BscI in *B. bronchiseptica*.** As mentioned above, our results suggested that Bcr4 interacts with BscI (Fig. 1). Because Bcr4 has various properties like those of chaperones for the type III secreted proteins (25), e.g., low molecular mass (18.1 kDa) and low isoelectric point (pI 4.41), we investigated whether Bcr4 is involved in the stability of BscI. We cultured the *B. bronchiseptica* wild-type, Bcr4-deficient strain (Δ*bcr4*, T3SS-inactive strain) (24), BspR-deficient strain (Δ*bspR*, T3SS-overproducing and -oversecreting strain) (14), or BspR- and Bcr4-deficient strain (Δ*bspR*, a T3SS-overproducing but -inactive strain) (24) in SS medium. The prepared WCL samples were separated by SDS-PAGE, and Western blotting was performed with antibodies against BscI, BscJ (SctJ, a protein composing the inner membrane ring of the T3SS), or Bcr4. As shown in Fig. S1, the negative regulatory function of BspR, which represses the *bsc* locus transcription, is enhanced in the strain that loses the activity of T3SS, e.g., the Δ*bcr4* mutant (24). Because of this property of BspR, the BscJ signal of the WCL sample was weaker in the Δ*bcr4* strain than in the wild type (Fig. 6A). In Δ*bspR* and Δ*bspR*Δ*bcr4* strains, the BscJ signals of the WCL samples were more strongly detected compared to the wild type (Fig. 6A) due to the BspR deficiency. On the other hand, the BscI signals of the WCL and CS samples were detected in the Δ*bspR* strain but not in the Δ*bspR*Δ*bcr4* strain (Fig. 6A). To investigate the presence of *bscI* mRNA in the Δ*bspR*Δ*bcr4* strain, a quantitative RT-PCR was performed. The results showed that the amount of *bscI* mRNA in the Δ*bspR*Δ*bcr4* strain was 1.5-fold higher than that of *bscI* mRNA in the Δ*bspR* strain (see Fig. S4), demonstrating that the *bscI* gene is transcribed in the Δ*bspR*Δ*bcr4* strain. These results suggest that Bcr4 is necessary for the stability of BscI in *B. bronchiseptica*. Finally, we examined whether Bcr4 binds to BscI inside of the *B. bronchiseptica* cells. As discussed below, we were unable to detect an interaction of Bcr4 with BscI by immunoprecipitation because of the low level of BscI protein in bacterial cells. Therefore, we attempted to reveal the interaction by using a cross linker. We treated the wild-type, Δ*bspR*, and Δ*bspR*Δ*bcr4* strains with disuccinimidyl suberate (DSS) and then prepared the WCL samples. The samples were separated by SDS-PAGE and then analyzed by Western blotting with anti-Bcr4 antibody. As a result, an extra signal of ~33 kDa was evident in the Δ*bspR* strain but not in the Δ*bspR*Δ*bscI* strain (Fig. 6B). The size of the extra signal was almost equivalent to that

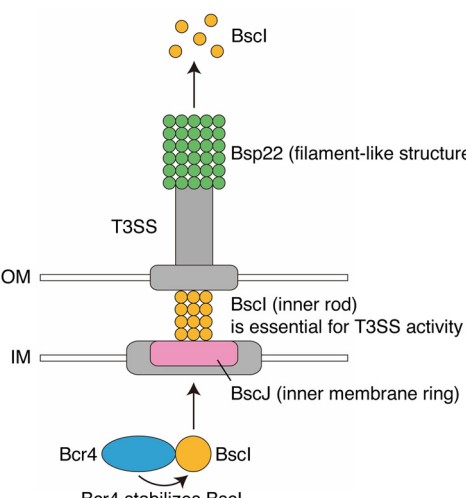

**FIG 7** Schematic depiction of the Bcr4 and BscI functions. In the *B. bronchiseptica* bacterial cytosol, Bcr4 (blue) stabilizes BscI (orange). The BscI is then presumed to be localized to the T3SS machinery (gray) and form the inner rod. BscI is essential for T3SS activity and secreted from the bacterial cell. BscJ (pink) is predicted to function as an inner membrane ring. Bsp22 (green) forms a filament-like structure. OM and IM, outer membrane (white) and inner membrane (white), respectively.

of the Bcr4-BscI complex, suggesting the possibility that Bcr4 binds to BscI in the *B. bronchiseptica* cells.

## DISCUSSION

In the present study, we found that deletion of BscI in *B. bronchiseptica* abolished the extracellular secretion of type III secreted proteins (Fig. 4 and 5). In addition, when Bcr4 was deleted in the *B. bronchiseptica* BspR-deficient strain, BscI (SctI, a protein composing the inner rod of the *Bordetella* T3SS) was not detected in the bacterial cells (Fig. 6A). These results suggest that Bcr4 contributes to the establishment of the T3SS in *B. bronchiseptica* by stabilizing BscI (Fig. 7).

To date, chaperones for needles, translocators, and effectors have been reported in bacteria that retain the T3SS, such as *Yersinia* (29–32). However, a chaperone for the inner rod protein has not been reported, and the mechanism underlying stability of the inner rod in the bacterial cell and the transport of inner rods to the T3SS remain unclear.

In the present study, it was suggested that both Bcr4 amino acid regions 58–109 and 110–173 are required for the binding of Bcr4 to BscI (Fig. 2B). To investigate whether Bcr4 amino acid region 1–57 is also required for the binding of Bcr4 to BscI, we attempted to purify Bcr4 lacking N-terminal amino acid region 1–57 (Bcr4Δ1-57-Strep). However, the amount of Bcr4Δ1-57-Strep produced in *E. coli* was extremely low (data not shown), so a pulldown assay could not be performed. Figure 3B suggests that Bcr4 amino acid region 159–163 is required for the function of the T3SS. However, it is still unclear whether Bcr4 amino acid region 159–163 is required for binding to BscI. Further analysis is needed to clarify whether the binding of Bcr4 to BscI is required for the function of T3SS.

In this study, the BopD signal was detected in the WCL of the *B. bronchiseptica* wild-type strain, but the BscI signal was not detected (Fig. 4A). We detected *bscI* mRNA in *B. bronchiseptica*, and the amount of *bscI* mRNA was not less than that of *bopD* mRNA (see Fig. S5). Therefore, the amount of BscI protein present in *B. bronchiseptica* was considered to be much lower than that of BopD. The reason for this finding is unknown, but it could be due to the low efficiency of translation of the *bscI* gene. In order to detect BscI in the WCL, we prepared the samples at 0 to 18 h after suspending the bacteria in liquid broth. However, no signals were detected in the WCL samples

**TABLE 1** Bacterial strains used in the study

| Strain | Relevant genotype | Reference or source |
|---|---|---|
| *B. bronchiseptica* | | |
| S798 | Wild-type strain | 6 |
| Δ*bscI* | BscI-deficient strain | This study |
| Δ*bscN* | BscN-deficient strain | 6 |
| Δ*bcr4* | Bcr4-deficient strain | 24 |
| Δ*bspR* | BspR-deficient strain | 14 |
| Δ*bspR*Δ*bscI* | BspR and BscI-deficient strain | This study |
| Δ*bspR*Δ*bcr4* | BspR- and Bcr4-deficient strain | 24 |
| | | |
| *E. coli* | | |
| DH10B | Host strain for pDONR201, p99-ccdB-V5, and pRK415-R4-R3-F | Invitrogen |
| BL21 | Host strain for pColdII | 35 |
| Sm10λ*pir* | Host strain for pABB-CRS2 | 6 |

prepared from the wild-type strain or the Δ*bcr4* strain (see Fig. S6). We also used the Δ*bspR* and Δ*bspR*Δ*bcr4* strains and detected BscI signals in the Δ*bspR* strain but not in the Δ*bspR*Δ*bcr4* double-knockout strain (see Fig. S6), findings strongly suggesting that Bcr4 is important for the stability of BscI.

In this study, we attempted to detect the BscI signal using a BspR-deficient strain (Δ*bspR*) in which the gene transcriptions on the *bsc* locus are promoted (Fig. 6), because the BscI signal was not detected in the WCL sample of the *B. bronchiseptica* wild-type (Fig. 4). The *bscJ* gene is located downstream of the *bscI* gene (Fig. 1A), and the BscJ protein is predicted to function as an inner membrane ring (SctJ), a component of the T3SS. The signal intensity of BscJ was not decreased in the WCL sample of the BspR/Bcr4 double-deficient strain (Δ*bspR*Δ*bcr4*) compared to that of the Δ*bspR* strain (Fig. 6A), suggesting that Bcr4 is not involved in the stability of BscJ. Therefore, it is suggested that Bcr4 specifically stabilizes BscI. In order to detect the interaction between Bcr4 and BscI in the *B. bronchiseptica* cells, immunoprecipitation was performed with anti-BscI antibody against Δ*bspR* lysate. The immunoprecipitated fractions were analyzed by Western blotting with anti-BscI antibody; however, an evident BscI signal was not detected (data not shown). The results of our immunoprecipitation assay were thus unable to demonstrate a specific interaction between Bcr4 and BscI. We speculate that *B. bronchiseptica* did not maintain a sufficient amount of BscI in WCL to detect the interaction by immunoprecipitation. As shown in Fig. 6B, we obtained an extra signal of ~33 kDa by Western blotting with anti-Bcr4 antibody when we analyzed the lysate of *B. bronchiseptica* treated with a cross-linker, DSS. When we performed the Western blotting with anti-BscI antibody for the cross-linked lysate, the extra signal was not detected (data not shown).

In order to examine whether Bcr4 has structural similarity to any chaperones for the type III secreted proteins produced by other bacteria, we used AlphaFold2. As a result, we detected significant structural similarities to the other chaperones, *Aeromonas* AcrH (33) and *Pseudomonas* PscG (34) (see Fig. S7). Although we obtained no plausible structural model for the interaction between BscI and Bcr4, the results strongly suggest that Bcr4 is a chaperone.

It is still unclear how Bcr4 antagonizes the BspR-negative regulation activity and how overexpression of Bcr4 promotes the production and secretion of type III secreted proteins in *B. bronchiseptica*. Further analysis is needed to elucidate the detailed molecular mechanism by which Bcr4 contributes to the regulation and establishment of the T3SS.

## MATERIALS AND METHODS

**Bacterial strains and cell culture.** The strains used in this study are listed in Table 1. *B. bronchiseptica* S798 (6) was used as the wild-type strain. The Δ*bscN* strain (a BscN-deficient strain), the Δ*bcr4* strain

(a Bcr4-deficient strain), the Δ*bspR* strain (a BspR-deficient strain), and the Δ*bspR*Δ*bcr4* strain (a BspR and Bcr4-deficient strain) have been described elsewhere (6, 14, 24). *E. coli* DH10B, BL21 (35), and Sm10λ*pir* were used for DNA cloning, recombinant protein expression, and conjugation, respectively (Table 1). *B. bronchiseptica* was grown on Bordet-Gengou agar plates at 37°C for 2 days. Fresh colonies of *B. bronchiseptica* were cultured in Stainer-Scholt (SS) medium (36) with a starting $A_{600}$ of 0.23 under vigorous shaking conditions at 37°C. L2 cells, a rat lung epithelial cell line (ATCC CCL-149), were maintained in F-12K (Invitrogen). The cell culture medium was supplemented with 10% fetal calf serum. L2 cells were grown at 37°C under a 5% $CO_2$ atmosphere.

**Preparation of proteins from culture supernatants and whole bacterial cell lysates.** Proteins secreted into bacterial culture supernatants and whole bacterial cell lysates were prepared as described previously (37). The loaded sample amounts were adjusted by the $A_{600}$ of each bacterial culture to load samples prepared from the same amounts of bacteria. The protein samples were separated by SDS-PAGE and analyzed by Western blotting.

**Antibodies.** Anti-BteA, anti-BopD, anti-Bsp22, BspR, BopB, and anti-Bcr4 antibodies were purified from rabbit serum in our previous study (6, 7, 10, 14, 24, 38). Mouse anti-V5 and anti-RNA polymerase beta subunit (RpoB) monoclonal antibodies were purchased from Santa Cruz Biotechnology and BioLegend, respectively. To detect filamentous hemagglutinin (FHA) signals, we used mouse anti-FHA serum kindly provided by K. Kamachi (National Institute of Infectious Diseases). To prepare the anti-BscI and anti-BscJ antibodies, the peptides corresponding to the C-terminal regions of BscI (CKAIGRATQNVDTLARMS) and BscJ (CRGEGRGGAGAGATEGAGHD) were conjugated with hemocyanin from keyhole limpets (Sigma), respectively, by using 3-maleimidobenzoic acid *N*-hydroxysuccinimide ester (Sigma). These cross-linked peptides were used to immunize rabbits, and the resulting antisera were incubated with the same peptides immobilized on epoxy-activated Sepharose 6B (Amersham) to obtain specific Ig fractions.

**Immunofluorescence staining of L2 cells infected with *B. bronchiseptica*.** Immunofluorescent staining assay was performed as previously described (37), with slight modifications. Briefly, L2 cells were seeded on coverslips in 6-well plates and incubated overnight. In order to detect adequate amounts of Bsp22 signals, we infected the L2 cells with *B. bronchiseptica* cultured in iron-depleted SS medium at an MOI of 2,000. To avoid excessive killing of L2 cells with bacteria, the plates were not centrifuged. After incubation for 1 h at 37°C under a 5% $CO_2$ atmosphere, the infected L2 cells were immunostained. The average numbers of Bsp22 signals on single L2 cells were measured by using a fluorescence microscope.

**LDH assays.** LDH assays were performed as previously described (37). Briefly, to examine whether LDH is released from *B. bronchiseptica*-infected cells, $5.0 \times 10^4$ L2 cells/well seeded in 24-well plates were infected with bacteria at an MOI of 50 or 200. The plates were centrifuged at $900 \times g$ for 5 min and incubated for 3 h at 37°C under a 5% $CO_2$ atmosphere. The amounts of LDH were measured spectrophotometrically using a Cyto-Tox 96 nonradioactive cytotoxicity assay kit (Promega). The LDH (%) level is shown as a ratio, with the value obtained from the well treated with 0.1% Triton X-100 set as 100%.

**Cross-linking assay.** The cross-linking assay was performed as previously described (39). A 100-$\mu$L culture of *B. bronchiseptica* was centrifuged at $20,000 \times g$, and the supernatant was removed. The pellet was washed with phosphate-buffered saline (PBS) and then resuspended in 100 $\mu$L of PBS. Next, disuccinimidyl suberate (DSS [Thermo]) dissolved in dimethyl sulfoxide was added at a final concentration of 10 mM. After incubation on ice for 1 h, Tris-HCl (pH 8.0) was added at a final concentration of 50 mM. The solution was centrifuged at $20,000 \times g$, and the supernatant was removed. The pellet was dissolved in 2× SDS-PAGE sample buffer. The samples were prepared from the same amounts of bacteria based on the $A_{600}$ values of the bacterial culture, separated by SDS-PAGE, and analyzed by Western blotting.

**Statistical analyses.** The statistical analyses were performed using a nonparametric test with a two-tailed *P* value with Prism v5.0f software (GraphPad, La Jolla, CA). *P* values of <0.05 were considered significant.

**Data availability.** The pulldown assay, the construction of *bscI* gene-disrupted or *bspR bscI* double strains, the construction of the plasmids used to produce Bcr4 derivatives and BscI complementation, the quantitative RT-PCR procedure, and protein structure prediction are described in Text S1 in the supplemental material. The plasmids and primers used in this study are listed in Tables S2 and S3, respectively.

## SUPPLEMENTAL MATERIAL

Supplemental material is available online only.

**SUPPLEMENTAL FILE 1**, PDF file, 1.7 MB.

## ACKNOWLEDGMENTS

This study was supported in part by grants (19K07542 to A.A., 20K07485 to A.K., and 19K07561 to T.H.) from the Ministry of Education, Culture, Sports, Science, and Technology and the Japan Society for the Promotion of Science (KAKENHI) and by a grant (JP22gm1610003 to M.S. and A.K.) from the Japan Agency for Medical Research and Development (AMED).

The funders had no role in the study design, date collection or analysis, the decision to publish, or the preparation of manuscript.

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
