## [Reviewer comments · Microbiology Spectrum]

Microbiology Spectrum

Bcr4 is a Chaperone for the Inner Rod Protein in the *Bordetella* Type III Secretion System

Masataka Goto, Akio Abe, Tomoko Hanawa, Masato Suzuki, and Asaomi Kuwae

Corresponding Author(s): Asaomi Kuwae, Kitasato University

Review Timeline:

Submission Date:	April 27, 2022
Editorial Decision:	May 30, 2022
Revision Received:	July 23, 2022
Editorial Decision:	August 5, 2022
Revision Received:	August 9, 2022
Accepted:	August 15, 2022

Editor: Eric Cascales

Reviewer(s): Disclosure of reviewer identity is with reference to reviewer comments included in decision letter(s). The following individuals involved in review of your submission have agreed to reveal their identity: Zongmin Du (Reviewer #1); Julien Bergeron (Reviewer #2)

Transaction Report:

DOI: <https://doi.org/10.1128/spectrum.01443-22>

May 30, 2022

Dr. Asaomi Kuwae
Kitasato University
Tokyo
Japan

Re: Spectrum01443-22 (Bcr4 is a Chaperone for the Inner Rod Protein in the *Bordetella* Type III Secretion System)

Dear Dr. Asaomi Kuwae:

Thank you for submitting your manuscript to Microbiology Spectrum. It has been sent to external reviewing, and I have now completed the reviewing process. As you will see in the reviewer's comments pasted below, the two referees are mainly positive with your work but they raised significant concerns that need to be addressed in a revised version. Notably, they suggest editing of some part of the manuscript (discussion) and require additional experiments to support some of the conclusions, notably by providing a Bcr4 structural model and by testing whether Bscl is stabilized by Bcr4. I therefore encourage you to carefully address the reviewer's comments and invite you to submit a revised version of your work. When submitting the revised version of your paper, please provide (1) point-by-point responses to the issues raised by the reviewers as file type "Response to Reviewers," not in your cover letter, and (2) a PDF file that indicates the changes from the original submission (by highlighting or underlining the changes) as file type "Marked Up Manuscript - For Review Only". Please use this link to submit your revised manuscript - we strongly recommend that you submit your paper within the next 60 days or reach out to me. Detailed instructions on submitting your revised paper are below.

Link Not Available

Sincerely,

Eric Cascales

Journals Department
Reviewer comments:

Reviewer #1 (Comments for the Author):

The authors report a previously uncharacterized protein Bcr4 in *B. bronchiseptica*, a zoonotic pathogen that causes respiratory infection. In vitro pull down assay was used to probe the interactions between Bscl and the various truncated Bcr4, and the results showed that C-termini of Bcr4 is essential for the binding with Bscl. By using the in vivo crosslinking assay, the authors showed that Bcr4 and Bscl can form a Bcr4-Bscl complex in a bspR mutant, although it was hard to be detected in the wild-type

strain. LDH cytotoxicity and fluorescence microscopy analysis results also confirmed the virulence phenotype of the mutant strains with the deletions of various region of Bcr4 C-termini. Based on these findings, the authors concluded that Bcr4 interacts with the rod protein of BscI and stabilize the BscI from premature degradation.

Major points:

1. The results presentation of Fig. 4 are confused and should be improved with additional explanations. Line 187: Why no signal of BscI was detected in the WCL samples of all strains, even in the wild-type strain that can express BscI? If the bacteria did not express these proteins, how did they be secreted into the culture supernatant? Why were Bsp22 present only in the CS but not in the WCL in Fig. 4 A?
2. Are there any homologs of Bcr4 in different *Bordetella* species? If present, what are the identities between Bcr4 and those homologs? These information should be introduced somewhere and is important for the reader to figure out to what extent the conclusion drawn by this work can be extended to the other bacteria.
3. Fig. 6: The crosslinking results between Bcr4 and BscI are convinced; however, the stabilization of BscI by binding with Bcr4 has not been fully demonstrated and additional data are needed to support this conclusion. A cycloheximide chase experiment of BscI in strains with the wild-type strain and the bcr4 mutant could be help.
4. Line 50: "the deletion of Bcr4 led to BscI instability in *B. bronchiseptica*", I cannot find the experimental results supporting this conclusion.
5. The discussion section should be more focused and concise.
6. Need some editing for grammar, particularly in the Discussion.

Minor points :

1. Lines 110-111 : I don't think based on the above description, one can conclude that Bcr4 will certainly interact with one of the proteins among BscI, BscJ, or BscK.
2. Line 124 : Why BteA N-terminal 1-312 amino acids region was used as a negative control should be introduced.
3. Lines 138-140: The conclusion that the 58-173 amino acids region is required for the interaction is not fully supported by the results. The author has not used a truncant of Bcr4-57-173, it is possible that Bcr4-57-173 cannot bind to BscI.
4. In the section of "The C-terminal region of Bcr4 is required for T3SS activity": The authors have not explained why they analyzed only the 158-173 residues in the LDH release assay. Based on the results in Fig. 2, the authors concluded that the 58-173 amino acids region is required for the interaction with BscI.
5. Line 212: Fig. 7 should be Fig. 5.
6. Fig. 4: The non-specific bands appearing in WCL and CS sample showed different locations, are they really the non-specific bands?

Reviewer #2 (Comments for the Author):

In this manuscript, Goto and co-workers report the characterisation of the protein Bcr4, present in the T3SS operon of *Bordetella*. The authors had previously demonstrated that this protein is essential for T3SS effector secretion, and here they further show that it is in fact a chaperone for the rod component BscI. This is a very intriguing observation, as this component does not have a dedicated chaperone in other bacterial species. Overall the manuscript is clear and well written, and the experiments are mostly well performed. However, a few elements require some clarification and additional analyses to make this manuscript easier to follow and fully convincing.

Specifically, the following need to be added to the manuscript:

- The T3SS nomenclature is very messy, and it makes this story sometimes hard to follow. A reminder in the introduction (and/or a table in the supplementary data) of the name of each components in the unified nomenclature, as proposed by Wagner and Diepold, would make it easier for the reader.
- An obvious omission here is a structural modelling analysis of Bcr4. A structural model, generated via AlphaFold, or even 2D prediction would go a long way to convince that even if it doesn't share any sequence identity to known T3SS chaperones, this protein belongs to one of the known chaperone classes. In addition, it would help understand how the boundaries for the various constructs (Figures 2A, 3A) were designed, which currently appears to have been done rather randomly.
- The use of sequence identity to associate the roles of BscF and BscI (line 84-85) is not really relevant, as the sequence conservation is really low. Secondary structure prediction, as well as the overall structure of the operon, is probably more important here.
- I am a bit concerned about what the authors have labeled as "non-specific signal" in figure 4A. This runs exactly at the same molecular weight as BscI-Strep, so I strongly suspect that this originated from some contamination. This experiment needs to be repeated properly, and if there is really a contamination band at this position, it must be investigated.

Staff Comments:

Preparing Revision Guidelines

Please return the manuscript within 60 days; if you cannot complete the modification within this time period, please contact me. If you do not wish to modify the manuscript and prefer to submit it to another journal, please notify me of your decision immediately so that the manuscript may be formally withdrawn from consideration by Microbiology Spectrum.

Comments for the Author

The authors report a previously uncharacterized protein Bcr4 in *B. bronchiseptica*, a zoonotic pathogen that causes respiratory infection. In vitro pull down assay was used to probe the interactions between BscI and the various truncated Bcr4, and the results showed that C-termini of Bcr4 is essential for the binding with BscI. By using the in vivo crosslinking assay, the authors showed that Bcr4 and BscI can form a Bcr4-BscI complex in a bspR mutant, although it was hard to be detected in the wild-type strain. LDH cytotoxicity and fluorescence microscopy analysis results also confirmed the virulence phenotype of the mutant strains with the deletions of various region of Bcr4 C-termini. Based on these findings, the authors concluded that Bcr4 interacts with the rod protein of BscI and stabilize the BscI from premature degradation.

Major points:

1. The results presentation of Fig. 4 is confused and should be improved with additional explanations. Line 187: Why no signal of BscI was detected in the WCL samples of all strains, even in the wild-type strain that can express BscI? If the bacteria did not express these proteins, how did they be secreted into the culture supernatant? Why were Bsp22 present only in the CS but not in the WCL in Fig. 4 A?
2. Are there any homologs of Bcr4 in different *Bordetella* species? If present, what are the identities between Bcr4 and those homologs? These information should be introduced somewhere and is important for the reader to figure out to what extent the conclusion drawn by this work can be extended to the other bacteria.
3. Fig. 6: The crosslinking results between Bcr4 and BscI are convinced;

however, the stabilization of BscI by binding with Bcr4 has not been fully demonstrated and additional data are needed to support this conclusion. A cycloheximide chase experiment of BscI in strains with the wild-type strain and the bcr4 mutant could be help.

4. Line 50: "the deletion of Bcr4 led to BscI instability in *B. bronchiseptica*", I cannot find the experimental results supporting this conclusion.
5. The discussion section should be more focused and concise.
6. Need some editing for grammar, particularly in the Discussion.

Minor points :

1. Lines 110-111 : I don't think based on the above description, one can conclude that Bcr4 will certainly interact with one of the proteins among BscI, BscJ, or BscK.
2. Line 124 : Why BteA N-terminal 1–312 amino acids region was used as a negative control should be introduced.
3. Lines 138-140: The conclusion that the 58-173 amino acids region is required for the interaction is not fully supported by the results. The author has not used a truncant of Bcr4-57-173, it is possible that Bcr4-57-173 cannot bind to BscI.
4. In the section of "The C-terminal region of Bcr4 is required for T3SS activity": The authors have not explained why they analyzed only the 158-173 residues in the LDH release assay. Based on the results in Fig. 2, the authors concluded that the 58-173 amino acids region is required for the interaction with BscI.
5. Line 212: Fig. 7 should be Fig. 5.
6. Fig. 4: The non-specific bands appearing in WCL and CS sample showed

different locations, are they really the non-specific bands?

Re: Spectrum01443-22 (Bcr4 is a Chaperone for the Inner Rod Protein in the *Bordetella* Type III Secretion System)

Reviewer #1 (Comments for the Author):

Major points:

Q1. The results presentation of Fig. 4 are confused and should be improved with additional explanations. Line 187: Why no signal of Bscl was detected in the WCL samples of all strains, even in the wild-type strain that can express Bscl? If the bacteria did not express these proteins, how did they be secreted into the culture supernatant? Why were Bsp22 present only in the CS but not in the WCL in Fig. 4 A?

Response We speculate that the amount of Bscl in the WCL sample was too low for detection by western blot. The bscl mRNA was detected (Fig. S5) and the Bscl signal were also detected in the supernatant sample (Fig. 4), suggesting the wild-type strain does produce Bscl. As shown in Fig. 4A, Bsp22 signals were detected in both the WCL and CS samples prepared from the wild-type and the Δ bscl+bscl strain.

Q2. Are there any homologs of Bcr4 in different *Bordetella* species? If present, what are the identities between Bcr4 and those homologs? These information should be introduced somewhere and is important for the reader to figure out to what extent the conclusion drawn by this work can be extended to the other bacteria.

Response As the reviewer suggested, we added an alignment of Bcr4 in *Bordetella* species shown in Fig. S2. We also added the following description in the Results: "Bordetella Bcr4 is highly conserved among *B. pertussis*, *B. parapertussis*, and *B. bronchiseptica* (Fig. S2)". (Page 5; lines 99 – 100)

Q3. Fig. 6: The crosslinking results between Bcr4 and Bscl are convinced; however, the stabilization of Bscl by binding with Bcr4 has not been fully demonstrated and additional data are needed to support this conclusion. A cycloheximide chase experiment of Bscl in strains with the wild-type strain and the bcr4 mutant could be help.

Response As the reviewer suggested, BscI degradation time course analysis is a suitable experiment to demonstrate the stabilization of BscI by Bcr4. Before using a translation inhibitor (cycloheximide is useful for eukaryotes), we first decided to prepare whole cell lysate samples at 0 hr to 18 hr after suspension of the bacteria in liquid broth. Unfortunately, no signals were detected in the whole cell lysate samples prepared from the wild-type strain and the *bcr4* mutant, suggesting that experiments using translation inhibitors would not be helpful in this case. We also used the *bspR* mutant and *bspR/bcr4* double mutant, and detected BscI signals in the *bspR* mutant, but not in the *bspR/bcr4* double mutant, strongly suggesting that Bcr4 is important for the stability of BscI. We added the results of these time course experiments as Fig. S6 and added the following description in the Discussion: "In order to detect BscI in the whole cell lysate, we prepared the samples at 0 hr to 18 hr after suspending the bacteria in liquid broth. However, no signals were detected in the whole cell lysate samples prepared from the wild-type strain or the $\Delta bcr4$ strain (Fig. S6). We also used the $\Delta bspR$ and $\Delta bspR\Delta bcr4$ strains, and detected BscI signals in the $\Delta bspR$ strain, but not in the $\Delta bspR\Delta bcr4$ double knockout strain (Fig. S6), strongly suggesting that Bcr4 is important for the stability of BscI". (Page 16; lines 309 – 314)

Q4. Line 50: "the deletion of Bcr4 led to BscI instability in *B. bronchiseptica*", I cannot find the experimental results supporting this conclusion.

Response We appreciate the reviewer's comment. We demonstrated that the BscI signal was clearly detected in the whole cell lysate sample prepared from $\Delta bspR$, but not from $\Delta bspR\Delta bcr4$ (Fig. 6A). To prevent any ambiguity, we rewrote the indicated sentence as follows: "Furthermore, in a *B. bronchiseptica* strain that overproduces T3SS component proteins, Bcr4 is required to maintain BscI in bacterial cells". (Page 3; lines 49 – 50)

Q5. The discussion section should be more focused and concise.

Response As the reviewer suggested, we omitted redundant sentences in the Discussion.

Q6. Need some editing for grammar, particularly in the Discussion.

Response We had the entire manuscript reviewed by a professional English proofreading company.

Minor points:

Q1. Lines 110-111: I don't think based on the above description, one can conclude that Bcr4 will certainly interact with one of the proteins among BscI, BscJ, or BscK.

Response We deleted the sentence the reviewer indicated.

Q2. Line 124: Why BteA N-terminal 1-312 amino acids region was used as a negative control should be introduced.

Response We used the BteA N-terminal region because the BteA is one of the proteins secreted from T3SS. In addition, it has been reported that a chaperone protein, BtcA, interacts with BteA from another group. This report described that BtcA interacted with the N-terminal region of BteA. Therefore, we thought the N-terminal region of BteA would be a suitable control to show the specificity of interaction between a protein and the protein secreted from the type III secretion system. We added the following sentence to the Results: "BteA is a protein secreted from the type III secretion system and interacts with its cognate chaperone BtcA through the N-terminal (9)". (Page 7, lines 126 – 127)

Q3. Lines 138-140: The conclusion that the 58-173 amino acids region is required for the interaction is not fully supported by the results. The author has not used a truncant of Bcr4-57-173, it is possible that Bcr4-57-173 cannot bind to BscI.

Response We rewrote the sentence as suggested: "Although it is still unknown which region of Bcr4 directly interacts with BscI, our results strongly suggest that both the Bcr4-58-109 and Bcr4-110-173 regions are required for the interaction". (Pages 7 – 8, lines 140 – 142)

Q4. In the section of "The C-terminal region of Bcr4 is required for T3SS activity": The authors have not explained why they analyzed only the 158-173 residues in the LDH release assay. Based on the results in Fig. 2, the authors concluded that the 58-173 amino acids region is required for the interaction with Bscl.

Response As the reviewer recommended, we attempted to create *B. bronchiseptica* strains that produce shorter Bcr4, but the amounts of shorter Bcr4 were very low when compared to that of the wild-type strain. We added the following description to the Discussion: "We attempted to create *B. bronchiseptica* strains that produce shorter Bcr4, e.g. Bcr4-1-57, 58-109, and 110-173, however, those truncated Bcr4 were produced at very low levels. Therefore, we were unable to evaluate whether or not these truncated proteins were functional in the *B. bronchiseptica*". (Pages 15 – 16; lines 298 – 302)

Q5. Line 212: Fig. 7 should be Fig. 5.

Response Thank you for pointing this out. We corrected "Fig. 7" to "Fig. 5". (Page 12; lines 221)

Q6. Fig. 4: The non-specific bands appearing in WCL and CS sample showed different locations, are they really the non-specific bands?

Response In order to demonstrate whether the signal is non-specific or not, we used the Δ bsp22 strain. We prepared supernatant fractions from the wild-type strain and Δ bsp22 strain, and carried out a western blot using anti-Bscl antibody. As shown in Fig. S3, the signal around 20 kDa disappeared in the Δ bsp22 sample, suggesting that the signal is a non-specific interaction between anti-Bscl antibody and an excess amount of Bsp22 on the membrane. We added the following passage to the Results: "In order to examine whether the signal we obtained around 20 kDa in the western blot using anti-Bscl antibody in Fig. 4A was specific or nonspecific, we prepared the supernatant fraction from Δ bsp22 strain. As a result, the signal disappeared in the supernatant fraction of the Δ bsp22 strain (Fig. S3), suggesting that the signal around 20 kDa obtained in the western blot was a nonspecific interaction between anti-Bscl antibody and

an excess amount of Bsp22 on the membrane". (Page 10; lines 192 – 197)

Reviewer #2 (Comments for the Author):

Q1. The T3SS nomenclature is very messy, and it makes this story sometimes hard to follow. A reminder in the introduction (and/or a table in the supplementary data) of the name of each components in the unified nomenclature, as proposed by Wagner and Diepold, would make it easier for the reader.

Response Thank you for pointing this out. We inserted the name of each component in the unified nomenclature in the Introduction. We also added a supplementary table (Table S1) describing the names and functions of the components.

Q2. An obvious omission here is a structural modelling analysis of Bcr4. A structural model, generated via AlphaFold, or even 2D prediction would go a long way to convince that even if it doesn't share any sequence identity to known T3SS chaperones, this protein belongs to one of the known chaperone classes. In addition, it would help understand how the boundaries for the various constructs (Figures 2A, 3A) were designed, which currently appears to have been done rather randomly.

Response As the reviewer suggested, we attempted to predict the Bcr4 structure by AlphaFold2. The results suggested a structural similarity of Bcr4 to a chaperone of the type III secreted protein. We added the following passage to the Discussion: "In order to examine whether Bcr4 has structural similarity to any chaperones for the type III secreted proteins produced by other bacteria, we used AlphaFold2. As a result, we detected significant structural similarities to the other chaperones, *Aeromonas* AcrH and *Pseudomonas* PscG (Fig. S7). Although we obtained no plausible structural model between Bcr4 and BscI, the results strongly suggest that Bcr4 is a chaperone". (Page 17; lines 334 – 339)

Q3. The use of sequence identity to associate the roles of BscF and BscI

(line 84-85) is not really relevant, as the sequence conservation is really low. Secondary structure prediction, as well as the overall structure of the operon, is probably more important here.

Response In keeping with the reviewer's suggestion, we changed the sentence to the following: "According to secondary structure prediction—e.g., the predicted positions of helix, and the overall structure of the operon—the BscF and BscI of *B. bronchiseptica* correspond to *Yersinia* needle YscF (SctF) and inner rod YscI (SctI), respectively". (Page 5; lines 85 – 88)

Q4. I am a bit concerned about what the authors have labeled as "non-specific signal" in figure 4A. This runs exactly at the same molecular weight as BscI-Strep, so I strongly suspect that this originated from some contamination. This experiment needs to be repeated properly, and if there is really a contamination band at this position, it must be investigated.

Response In order to determine whether the signal is non-specific, we used the Δ bsp22 strain. We prepared supernatant fractions from the wild-type strain and Δ bsp22 strain, and carried out a western blot using anti-BscI antibody. As shown in Fig. S3, the signal around 20 kDa disappeared in the Δ bsp22 sample, suggesting that this signal is a non-specific interaction between anti-BscI antibody and an excess amount of Bsp22 on the membrane. We added the following explanatory sentences to the Results: "In order to examine whether the signal we obtained around 20 kDa in the western blot using anti-BscI antibody in Fig. 4A was specific or nonspecific, we prepared the supernatant fraction from Δ bsp22 strain. As a result, the signal disappeared in the supernatant fraction of the Δ bsp22 strain (Fig. S3), suggesting that the signal around 20 kDa obtained in the western blot was a nonspecific interaction between anti-BscI antibody and an excess amount of Bsp22 on the membrane". (Page 10; lines 192 – 197)

August 5, 2022

Dr. Asaomi Kuwae
Kitasato University
Tokyo
Japan

Re: Spectrum01443-22R1 (Bcr4 is a Chaperone for the Inner Rod Protein in the *Bordetella* Type III Secretion System)

Dear Dr. Asaomi Kuwae:

Thank you for submitting your revised manuscript to Microbiology Spectrum. It has been sent back to the two original reviewers. As you will see, while reviewer #2 appreciated your responses, reviewer #1 is still critical and has several questions that need to be addressed in a second revision. I therefore encourage you to carefully address these issues and to resubmit a second revised version of your work. When submitting the revised version of your paper, please provide (1) point-by-point responses to the issues raised by the reviewers as file type "Response to Reviewers," not in your cover letter, and (2) a PDF file that indicates the changes from the original submission (by highlighting or underlining the changes) as file type "Marked Up Manuscript - For Review Only". Please use this link to submit your revised manuscript - we strongly recommend that you submit your paper within the next 60 days or reach out to me. Detailed instructions on submitting your revised paper are below.

Link Not Available

Sincerely,

Eric Cascales

Journals Department
Reviewer comments:

Reviewer #1 (Comments for the Author):

The authors have attempted to address my previous concerns and the current version of manuscript is much better. However, there still are some of concerns need to be addressed.

1. The amount of BscI in lysate of the wild type bacteria was too low to be detected by western blot. Thus, the authors analyzed the transcripts of bscI gene in the WT strain. Fig. S5, The q-PCR results shown in is hard to understand. What is the reference gene and what does the fold represent? I wonder whether the level of bscI mRNA in a bcr4 mutant is lower than that in the wild-type bacteria?
2. Fig. 6, the higher level of BscI in the bspR mutant than that in the bspR/bcr4 double mutant does not definitely mean a higher stability of BscI in the presence of Bcr4, since this phenomena could also result from an elevated expression of BscI. The

authors have neither discussed this point nor precluded this possibility by providing supporting evidence.

3. Fig.2, the title should be reorganized , because the regions required for the binding of Bcr4 to BscI have not been identified as claimed by the current title.

4. Fig.5, the authors have not explained why they analyzed only the 158-173 residues in the LDH release assay in the section of "The C-terminal region of Bcr4 is required for T3SS activity", but mentioned this issue in the discussion section. To make the story easier to follow, the explanation should be added to the section of result rather than discussion.

Reviewer #2 (Comments for the Author):

The authors have modified the manuscript according to both reviewers' request, and have notably added several experimental controls, and further sequence analyses.

As a consequence, this manuscript is significantly improved, and I am delighted to recommend its publication in Spectrum.

Staff Comments:

Preparing Revision Guidelines

Please return the manuscript within 60 days; if you cannot complete the modification within this time period, please contact me. If you do not wish to modify the manuscript and prefer to submit it to another journal, please notify me of your decision immediately so that the manuscript may be formally withdrawn from consideration by Microbiology Spectrum.

The authors have attempted to address my previous concerns and the current version of manuscript is much better. However, there still are some of concerns need to be addressed.

1. The amount of BscI in lysate of the wild type bacteria was too low to be detected by western blot. Thus, the authors analyzed the transcripts of bscI gene in the WT strain. Fig. S5 , The q-PCR results shown in is hard to understand. What is the reference gene and what does the fold represent? I wonder whether the level of bscI mRNA in a bcr4 mutant is lower than that in the wild-type bacteria?

2. Fig. 6, the higher level of BscI in the bspR mutant than that in the bspR/bcr4 double mutant does not definitely mean a higher stability of BscI in the presence of Bcr4, since this phenomena could also result from an elevated expression of BscI. The authors have neither discussed this point nor precluded this possibility by providing supporting evidence.

3. Fig.2, the title should be reorganized , because the regions required for the binding of Bcr4 to BscI have not been identified as claimed by the current title.

4. Fig.5, the authors have not explained why they analyzed only the 158-173 residues in the LDH release assay in the section of "The C-terminal region of Bcr4 is required for T3SS activity", but mentioned this issue in the discussion section. To make the story easier to follow, the explanation should be provided in the section of result rather than discussion.

Re: Spectrum01443-22R1 (Bcr4 is a Chaperone for the Inner Rod Protein in the *Bordetella* Type III Secretion System)

Reviewer #1 (Comments for the Author):

Q1. The amount of BscI in lysate of the wild type bacteria was too low to be detected by western blot. Thus, the authors analyzed the transcripts of bscI gene in the WT strain. Fig. S5, The q-PCR results shown in is hard to understand. What is the reference gene and what does the fold represent? I wonder whether the level of bscI mRNA in a bcr4 mutant is lower than that in the wild-type bacteria?

Response We have described the protocol of absolute quantification in the Supplementary Information Materials and Methods submitted on July 23. (Pages 10; Lines 183 – 187)

We added the following descriptions in the Fig. S5 legend: “The relative ratio of bscI mRNA is shown when the bopD mRNA amount is set as 1.”

As described in the Results section (Pages 11; lines 202 – 205), in a bcr4 mutant, transcription of the bsc locus is repressed. The bscI gene is included in bsc locus. Therefore, the level of bscI mRNA in a bcr4 mutant is supposed to be lower than that in the wild-type.

Q2. Fig. 6, the higher level of BscI in the bspR mutant than that in the bspR/bcr4 double mutant does not definitely mean a higher stability of BscI in the presence of Bcr4, since this phenomena could also result from an elevated expression of BscI. The authors have neither discussed this point nor precluded this possibility by providing supporting evidence.

Response We showed the expression (mRNA) levels in the bspR mutant and the bspR/bcr4 mutant in Fig. S4 and have described these results in the text (Page 13 – 14; lines 259 – 263).

Q3. Fig.2, the title should be reorganized, because the regions required for the binding of Bcr4 to BscI have not been identified as claimed by the current title.

Response We replaced the title with the following one: "Pull-down assays between BscI and truncated versions of Bcr4".

Q4. Fig.5, the authors have not explained why they analyzed only the 158-173 residues in the LDH release assay in the section of "The C-terminal region of Bcr4 is required for T3SS activity", but mentioned this issue in the discussion section. To make the story easier to follow, the explanation should be added to the section of result rather than discussion.

Response As the reviewer suggested, we move the following sentences to the Results (Page 9; line 168 –172) from the Discussion: "We attempted to create *B. bronchiseptica* strains that produce shorter Bcr4, e.g. amino acid regions 1-57, 58-109, and 110-173, however, those truncated Bcr4 were produced at very low levels (data not shown). Therefore, we were unable to evaluate whether or not these truncated proteins were functional in *B. bronchiseptica*".

August 15, 2022

Dr. Asaomi Kuwae
Kitasato University
Tokyo
Japan

Re: Spectrum01443-22R2 (Bcr4 is a Chaperone for the Inner Rod Protein in the *Bordetella* Type III Secretion System)

Dear Dr. Asaomi Kuwae:

Thank you for submitting your revision. I am please to accept your manuscript has been accepted, and I am forwarding it to the ASM Journals Department for publication. You will be notified when your proofs are ready to be viewed.

Sincerely,

Eric Cascales
Editor, Microbiology Spectrum

Journals Department
Supplemental file 1: Accept